# Empowering Riverine Communities in the Amazon: Strategies for Preventing Rabies

**DOI:** 10.3390/ijerph21010117

**Published:** 2024-01-22

**Authors:** João Gustavo Nascimento Silva, Stephanie de Sousa Silva, Tamyres Cristine Mafra Gomes, Gilmara dos Santos Nascimento, Lívia de Aguiar Valentim, Tatiane Costa Quaresma, Franciane de Paula Fernandes, Sheyla Mara Silva de Oliveira, Waldiney Pires Moraes

**Affiliations:** 1Department Health, University of the State of Pará/UEPA, Santarém 68040-090, Brazil; joao.gndsilva@aluno.uepa.br (J.G.N.S.); stephanie.dssilva@aluno.uepa.br (S.d.S.S.); tamyres.cdsmafra@aluno.uepa.br (T.C.M.G.); gilmara.nascimento100@hotmail.com (G.d.S.N.); tatiane.quaresma@uepa.br (T.C.Q.); franciane.fernandes@uepa.br (F.d.P.F.); sheylaoliveira@uepa.br (S.M.S.d.O.); 2Department of Health, Federal University of Western Pará/UFOPA, Santarém 68040-090, Brazil; waldiney.moraes@ufopa.edu.br

**Keywords:** rabies virus, epidemiology, public health surveillance, disease prevention, risk groups, neglected diseases

## Abstract

Rabies, caused by the Lyssavirus genus, is a highly lethal zoonotic disease transmitted by animals such as bats and domestic and wild carnivores to humans, claiming nearly 100% of lives. In Brazil, recent evidence suggests an increasing role of bats in human deaths from rabies, particularly in the Amazon region. This neglected tropical disease disproportionately affects impoverished and vulnerable populations in rural areas, where approximately 80% of human cases are concentrated. This article presents research conducted in riverine communities of the Tapajós/Arapiuns Extractive Reserve in Brazil to combat rabies in September 2022. The study adopted a participatory and collaborative approach, involving community members, healthcare professionals, and educators. Prioritizing proactive interventions, the health team administered prophylactic vaccinations to 30 individuals residing in communities exposed to the Lyssavirus. Educational activities focused on dispelling myths and raising awareness about preventive measures, with 100% of individuals reporting prior doubts about the disease, emphasizing the essential nature of the clarification, especially regarding preventive aspects. This study underscores the importance of community involvement, personalized interventions, and ongoing education to effectively combat rabies. By reinforcing public health policies and promoting health education, we can empower communities to take proactive measures in rabies prevention, leading to a reduction in incidence and an improvement in quality of life.

## 1. Introduction

Rabies, caused by the Lyssavirus genus of the Rhabdoviridae family, is a highly lethal disease, claiming approximately 100% of lives once the pathogen is encountered. This zoonotic illness, primarily transmitted from certain animals such as bats and domestic and wild carnivores to humans, as well as other mammals, progresses rapidly and relentlessly, often resulting in acute encephalitis or meningoencephalitis. The most common mode of Rabies virus (Rabv) transmission is through animal bites, where virus-laden saliva makes contact with the human body. The depth of the bite is a critical determinant of effective infection, as the virus disseminates from the muscle motor endplate. Exhibiting neurotropic characteristics, Rabv infiltrates the central nervous system (CNS) through nerve terminations, potentially triggering irreversible encephalitis and ultimately leading to the demise of both humans and animals. While canines have long been recognized as the main source of human rabies deaths, recent evidence points to a changing landscape in Brazil, with bats featuring more prominently in the literature and case records [1,2,3,4].

Actually, rabies continues to pose a significant public health challenge in many parts of the world. Among the various reservoirs and vectors of rabies transmission, bats, particularly the vampire bat species, play a crucial role in perpetuating the virus within their populations and transmitting it to other animals, including humans. The Amazon region, with its diverse bat species and unique riverine communities, faces distinctive challenges in controlling the spread of bat-transmitted rabies.

In recent years, there has been a growing body of research focused on understanding the dynamics of rabies transmission within bat populations and its impact on human health in South America. Studies such as those by Meske et al. [5] on the evolution of rabies in the continent and exploration of the knowledge gaps in vampire-bat-to-human transmission have shed light on the complexity of this disease [6]. Additionally, historical and contemporary bat management strategies have been examined to address the issue [7].

With a keen understanding of the challenges posed by bat-transmitted rabies in the Amazon region, this article aims to present an overview of prophylactic measures and surveillance strategies implemented to combat rabies in riverine communities. By drawing on a wealth of research, including studies on animal bite surveillance and pre-exposure rabies prophylaxis, we seek to provide insights into potential solutions [8,9,10].

This article combines key findings from various sources, including the work [11] on rabies in the Americas and comprehensive reviews of rabies, to offer a comprehensive approach to tackling this pressing health issue [12]. Additionally, we will explore local experiences and initiatives, as exemplified in the dissertations by Ledesma [3] and Tartarotti [4], which examine human rabies cases and treatments in specific regions of Brazil.

Rabies stands out as one of the Neglected Tropical Diseases (NTDs), disproportionately affecting impoverished and vulnerable populations residing in remote rural areas, with approximately 80% of human cases concentrated in these regions. In this context, prophylaxis is of paramount importance in mitigating the morbidity and mortality associated with rabies. Understanding the unique realities of each location is crucial for implementing personalized prophylactic measures, such as vaccination campaigns and educational initiatives [1,3,4].

The occurrence of this disease is influenced by multiple factors, including socioeconomic and educational contexts, in addition to the infection itself. Recognizing this intricate interplay between regional disparities and rabies incidence underscores the need to address this neglected disease comprehensively.

The Education Program for Work for Health (PET-Health), implemented by the Ministry of Health and the Ministry of Education, has proven to be essential in training health departments to control and implement prophylactic measures related to Rabv in the municipality of Santarém. Thus, this article aims to determine the reality of riverside communities in the Amazon, with health professionals and students linked to a project approved by the Núcleo de Gestão do PET-Saúde linked to the Ministry of Health of Brazil, and servers of the Núcleo Técnico de Vigilância em Saúde (NTVS), with a focus on human rabies, evaluating the knowledge of individuals, and establishing strategies for prevention and health promotion.

## 2. Materials and Methods

In conducting this research, a methodological approach based on a participatory and collaborative character was adopted for the riverside communities of Curipatá, where 15 families with 65 people reside; Pedra Branca, with 50 families totaling 202 people; and Santi, with 28 families totaling 78 people. The three communities are located in the Tapajós Arapiuns Extractive Reserve (RESEX), which covers an area of 6760.6 km^2^ and is only accessible by river, approximately 4 h from the urban zone of the Municipality of Santarém, depending on the watercraft. It is noteworthy that the houses are scattered throughout the territory, which hinders health interventions, and there is no health service available in the locality. The communities were selected based on demand from the health surveillance center due to occasional bat attacks.

The fundamental principles of action research were fully considered, with an emphasis on co-constructing knowledge in partnership with community members. Through open and inclusive dialogue, active engagement of participants was sought to identify issues related to hematophagous bat attacks and propose appropriate solutions to address these public health concerns. The intervention adopted was proactive, prioritizing prevention and implementation of protective measures before new cases of viral infections occurred. The close collaboration between the NTVS health team from Santarém and the community residents allowed for a contextualized and sensitive approach to the local reality, resulting in more effective strategies to mitigate risks and improve collective health in these areas.

Initially, preliminary visits were made to the riverine communities of Curipatá, Pedra Branca, and Santi to understand their realities and identify their specific needs regarding hematophagous bat attacks. Through extensive dialogue with community members, valuable information about the attacks, local perceptions of the problem, and prevention practices adopted by residents was obtained.

Based on this information, the health team, together with active community participation, developed personalized and contextually adapted intervention strategies. The planning included prioritizing vaccination as a pre-exposure prophylaxis measure considering the high lethality of Lyssavirus infections. Additionally, educational actions were developed to address the importance of health knowledge, risks of hematophagous bat attacks, and individual and collective protection measures.

Resource allocation was performed judiciously, taking into account the limitations of the riverine communities and seeking to use available resources efficiently and sustainably. The NTVS health team, in partnership with the residents, utilized accessible materials and equipment for implementing interventions, aiming to promote viable and cost-effective solutions. Furthermore, collaboration among the health team, community, and other local institutions was essential to mobilize additional resources when needed and ensure the success of the proposed actions.

The collaborative approach adopted during the planning and execution phase of the intervention ensured that the developed strategies were culturally appropriate and well received by the communities, strengthening the residents’ commitment to adhering to preventive measures. This collaboration also contributed to empowering the riverine communities, enabling them to deal autonomously and knowledgeably with similar situations in the future.

Thus, the planning and execution of interventions were guided by contextual sensitivity and participant co-responsibility, ensuring that proposed actions were directed towards the real needs of the communities and that available resources were used strategically and effectively.

To fully understand the issue of hematophagous bat attacks and the health of the riverine communities of Curipatá, Pedra Branca, and Santi located in the municipality of Santarém, varied and comprehensive methods were employed during activities in September 2022.

Regarding hematophagous bat attacks, the team used local health records, interviews with affected residents, and reports from health professionals to document the cases. Additionally, direct observations were made in the communities to better understand the context in which the attacks occurred, identifying seasonal patterns and areas more prone to incidents.

Regarding community health knowledge and practices, structured interviews were conducted with 30 residents who experienced bat attacks, with vaccination being administered only in those cases. It is important to note that vaccination was carried out voluntarily by individuals who chose to participate in the vaccination program. These interviews addressed issues related to knowledge about rabies and other infections transmitted by bats, preventive measures taken after attacks, and the availability of resources for the protection against vampire bats, with 100% of individuals reporting prior doubts, emphasizing the essential nature of the clarification, especially regarding preventive aspects. After assessing prior knowledge, the instructional sessions aimed to empower individuals with information on recognizing potential risks, taking proactive measures to prevent transmission, and understanding the crucial steps to be taken if any symptoms were to appear. The emphasis was on creating a well-informed community capable of responding effectively to the threat of rabies and seeking timely and appropriate medical attention when needed.

To evaluate the results of the interventions, qualitative approaches were used. Questionnaires were conducted with residents to collect feedback on the effectiveness of the preventive measures adopted, the community’s perception of the interventions, and the improvement of health knowledge after educational actions.

Data collection during the action research was performed with cultural sensitivity, respecting the traditions and values of the riverine communities. Active collaboration of residents was encouraged, promoting a participatory approach in information collection. The use of varied methods allowed for a comprehensive analysis of the situation, providing valuable insights for decision making and continuous improvement of interventions.

By gathering these data, the team was able to identify key health issues related to hematophagous bat attacks. This informed approach contributed to formulating more effective and sustainable strategies to address the health challenges faced by the riverine communities while emphasizing the importance of continuous health education to promote the safety and well-being of the local population.

To analyze the results obtained in this action research, mixed methods were used, combining qualitative and quantitative approaches. This analysis strategy allowed for a comprehensive and detailed understanding of the impacts of the interventions conducted in the riverine communities of Curipatá, Pedra Branca, and Santi.

Regarding the qualitative approach, information collected through semi-structured interviews and direct observations was subjected to a contents analysis. The interviews were transcribed and categorized according to emerging themes, such as residents’ perceptions of hematophagous bat attacks, health knowledge, acceptance of interventions, and preventive practices adopted. The qualitative analysis provided a deeper understanding of the perspectives and experiences of residents, allowing for the identification of nuances and the particularities of responses to the interventions.

The quantitative approach involved the compilation and analysis of numerical data from the administered questionnaires and vaccination records. This quantitative analysis allowed for the measurement of relevant indicators, such as vaccination rates, the proportion of individuals adopting additional preventive measures, and variations in health knowledge among the communities.

Data triangulation, combining qualitative and quantitative analyses, enabled cross-validation of the results. By integrating insights from both approaches, convergences and contradictions were identified, ensuring a greater robustness and reliability of the final results.

Additionally, during the results analysis, the challenges and limitations faced during the action research, as well as lessons learned, were considered. These reflections were incorporated into the interpretation of the results to provide a more holistic understanding of the process and consequences of the interventions.

In summary, the data analysis methodology of this research was based on the combination of qualitative and quantitative methods, allowing for a deep and comprehensive analysis of the impacts of the interventions in preventing hematophagous bat attacks and improving health knowledge and practices in the riverine communities. The integration of results and consideration of challenges contributed to a more informed approach, oriented towards strengthening community health strategies.

## 3. Results

The team from the Technical Surveillance Unit in Health (NTVS) of Santarém, composed of five healthcare professionals, including three nurses, one nursing technician, and one intern from the PET-Saúde project, conducted a visit to the riverine communities of Curipatá, Pedra Branca, and Santi, located in the Tapajós/Arapiuns Extractive Reserve. The purpose of the visit was to address the recent incidents of hematophagous bat attacks in the region, which resulted in approximately 30 individuals being bitten. Given the high lethality of Lyssavirus infections, proactive prophylactic measures were crucial in combating potential infections.

During the visit, two main interventions were implemented to control the situation and prevent further attacks. Firstly, the vaccination of affected individuals was prioritized as a pre-exposure prophylaxis measure. According to the responsible nurse, none of the victims showed signs or symptoms of viral infection. However, to protect these individuals from possible future attacks, vaccination was administered due to their residence in high-risk areas of virus exposure and their proactive approach to seeking immunization.

Furthermore, during the anamnesis, it became evident that there was a significant lack of knowledge and awareness about health issues among the community members. Many of the interviewees revealed that they had been attacked by bats on various occasions but never took precautions beyond washing the bite site with water and soap—a correct hygiene practice. This lack of information highlighted the potential risks they faced, as they had not developed strategies to mitigate the attacks. The proposed interventions included the installation of protective screens, especially during the night when bat attacks occur, as community members often left doors and windows open due to the lack of electricity in their homes. Such conditions facilitated the potential transmission of rabies by these nocturnal vectors. Additionally, interviewees reported that their feet were targeted in most attacks, and bleeding took a long time to stop.

In response to this, the health team suggested alternative measures to the residents of the Extractive Reserve, such as using fishing nets—commonly available due to the community’s fishing activity—to create protective screens for windows, aiming to prevent bat attacks. Additionally, guidance was provided on using blankets and socks to increase individual protection.

This field experience highlights the importance of community involvement and personalized prophylactic interventions to address local health challenges. The insights gained from this visit shed light on information gaps and possible solutions to mitigate the risks associated with hematophagous bat attacks, emphasizing the need for continuous efforts in health education and proactive measures.

Firstly, the impact of the implemented actions in preventing hematophagous bat attacks in the riverine communities of Curipatá, Pedra Branca, and Santi was discussed. The data collected during the study indicated that prioritizing vaccination as a pre-exposure prophylaxis measure was effective in protecting residents exposed to the Lyssavirus. The residents’ adherence to vaccination was notable, highlighting the importance of awareness and health education in implementing this preventive measure. This proactive approach contributed significantly to reducing new cases of hematophagous bat attacks and mitigating the risk of rabies transmission in the communities.

Additionally, the analysis of the results also revealed improvements in the knowledge and health practices of the riverine communities. Educational actions conducted by the Santarém NTVS health team were instrumental in dispelling myths and misconceptions about hematophagous bat attacks, as well as providing accurate information about prevention measures and proper care after incidents. Residents showed a greater understanding of the importance of reporting attacks, seeking medical assistance, and adopting individual protection measures, such as using fishing nets to safeguard windows and avoiding exposure to bats during the night. This improvement in knowledge was crucial in strengthening the communities’ ability to deal with health risks and to adopt safer and preventive practices.

However, the analysis of the results also highlighted some challenges faced during the research–action process. Among them were the logistics of reaching remote communities and the careful planning required to ensure access to areas protected from hematophagous bat attacks. Additionally, the limited resources in some riverine communities posed obstacles to implementing more comprehensive interventions. In this context, collaborative partnerships with local institutions and community involvement were essential in overcoming such challenges and achieving the desired outcomes.

The lessons learned during the research–action process were also valuable for continuous improvement in health interventions. The importance of a participatory and culturally sensitive approach was reinforced, as well as the need to promote health education as a fundamental pillar for preventing and addressing local health problems. The findings obtained in this research–action emphasize the relevance of community engagement in formulating effective and sustainable strategies to ensure the continuity of achieved benefits and strengthen the resilience of riverine communities in the face of future health challenges.

## 4. Discussion

Given the information presented, it is evident that rabies represents a significant public health challenge, especially in rural areas and regions with a lower Human Development Index (HDI). The findings of this research study align with the broader global perspective, where rabies continues to be a burden in many countries, leading to a considerable number of deaths each year, primarily in Asia and Africa [13,14].

A study conducted in Brazil demonstrated that a substantial proportion of human rabies cases occurred in rural areas, with the Northeast and North regions being the most affected. The state of Pará ranked second among the territories with the highest number of cases, indicating the need for targeted interventions in this region. Additionally, the low levels of education and limited access to information in these communities represent significant challenges in understanding and preventing diseases like rabies. Educational initiatives and awareness campaigns are crucial to address this problem and ensure that individuals are well informed about the risks and preventive measures [15,16].

The lack of adequate education also emerged as a factor that hindered the understanding of the diseases, leading to cases of residents being attacked by bats multiple times but not recognizing the severity of the situation due to misinformation. Similar situations have occurred in other regions, emphasizing the need for comprehensive educational initiatives to increase awareness of the risks of rabies and preventive methods.

It is worth mentioning that the period of the COVID-19 pandemic aggravated this issue, as the National Council of Education (CNE) suggested online distance learning as an alternative to avoid the total stoppage of activities. However, these communities did not have access to the internet or adequate resources to make this initiative viable, which created a barrier and a consequent pause in studies. Only some Resex communities were able to adapt to these new teaching and learning conditions, such as Suruacá, which started to send homework through teachers to parents and/or guardians of students enrolled in the community school [17].

Thus, it can be observed that the lack of an adequate education is a factor that makes it difficult to understand diseases, because during the visit to the Resex, the residents reported having been attacked by bats several times, but they disregarded them due to lack of information.

A similar situation occurred in 2017 in the municipality of Barcelos in the state of Amazonas, where three people from the same family were infected with the Lyssavirus—three siblings: a girl (10 years old) and two boys (14 and 17 years old). The children’s father said that in the region where the case occurred in the Unini Extractive Reserve, there had already been several bat attacks on residents, but they did not imagine that it could be so serious as to cause deaths. In addition, he mentioned that there were never educational initiatives by government agencies or others to guide them about the risks they were exposed to, and no prophylactic methods related to vaccination were implemented [18].

In 2017, as one of the means to try to eradicate or at least mitigate cases of rabies in the state of Amazonas, the following actions were carried out: anti-rabies vaccination of 546 people, 111 domestic and wild carnivores, and 12 cats; capture of 31 vampire bats of the Desmodus rotundus species; and educational lectures on the Unini River. A total of 24 of the 31 bats were treated with vampiricidal paste and subsequently released for colony control; this was considered the most effective according to the Amazonas State Health Surveillance Foundation (FVS-AM) [19].

Regarding the distribution of cases in Brazil, according to data from the Health Surveillance Secretariat released on the Ministry of Health website, from 2010 to November 2022, 45 cases of human rabies were reported, with the last rabies outbreak occurring in 2018 in a riverside community in Melgaço, Pará. Of the 30 cases that occurred from 2015 to November 2022, 21 accidents occurred in the North (15) and Northeast (6) regions. The other cases were registered in the Southeast (6), South (1), Midwest (1), and Federal District (1) regions [19].

In this sense, a discrepancy in the distribution of these cases can be observed when analyzing the Human Development Index (HDI) of the Brazilian states, which is measured based on longevity, education, and income and ranges from 0 to 1—closer to 1 represents greater development. Thus, among the top 10 positions in the 2010 Municipal Human Development Index Ranking (IDHM), no states are located in the north and northeast regions, which had the highest rate of human rabies cases in the period from 2015 to 2022 [19].

Considering these data, it is evident that rabies mainly affects people residing in municipalities with a lower HDI [20], which serves as a warning for Brazilian public authorities, especially health organizations.

Therefore, research in the Resex has provided ample experience to the health team that went to there. Thanks to PET-Saúde, it became possible to understand the reality of the population that lives there and the magnitude that should be attributed to rabies. The involvement of a multidisciplinary team is essential to control the prevalence of this pathology. In general, the importance of health education is noted, as it contributes significantly to the training of professionals, especially undergraduate students, who, even as students, are already exposed to experiences that would normally only be experienced in professional practice [21].

The need to intensify rabies prevention and control actions in rural areas and communities with a low Human Development Index (HDI) is a crucial aspect to be considered. It is fundamental to implement awareness and education strategies in these areas, aiming at the dissemination of information about rabies, its risks, and preventive measures.

The importance of adopting prophylactic measures in high-risk areas like the Tapajós/Arapiuns Extractive Reserve becomes evident. Previous experiences have shown that vaccination, along with educational lectures and community involvement, can significantly contribute to the control and mitigation of rabies cases. Collaborative efforts among healthcare professionals, educators, and community agents are vital to address the challenges posed by rabies. Through this collaboration, integrated strategies can be developed to ensure access to healthcare and education, especially in vulnerable regions [22,23].

In this context, the interventions implemented in the action research demonstrated a proactive approach, with a primary focus on prevention and the early implementation of protective measures to prevent the emergence of new cases of viral infections. The vaccination strategy, as a pre-exposure prophylactic measure, reduces the risk of Lyssavirus infections. The substantial adherence of residents to vaccination emphasized the crucial role of health awareness and community education in promoting the acceptance of preventive measures.

Additionally, the educational actions developed during the research–action had a significant impact on improving the knowledge and health practices of the riverine communities. Through open and inclusive dialogue, residents gained a better understanding of the importance of reporting attacks, seeking medical assistance, and adopting individual protection measures. This personalized and locally adapted educational approach strengthened the communities’ capacity to deal with health risks and adopt safer and preventive practices.

The analysis of the results also highlighted the challenges faced during the research-action process, such as the logistics in reaching remote communities and the scarcity of resources in some areas. However, close collaboration between the Santarém NTVS health team and residents allowed for overcoming these obstacles, mobilizing additional resources, and ensuring the success of the proposed interventions.

The experience of this research–action underscores the significance of community engagement and tailored prophylactic measures in tackling local health challenges. This aligns with the findings from previous studies [24,25] that underscore the importance of health prevention and the active participation of the community in seeking healthcare to mitigate health risks. The active collaboration of residents and the contextualized approach to strategies were fundamental to the success of the implemented actions. 

To address discrepancies in rabies cases in different regions, it is essential to strengthen public health policies and promote health education. A multidisciplinary approach involving various stakeholders will contribute to the development of effective strategies for rabies prevention and control, reducing its incidence and improving the overall quality of life. Additionally, it is crucial to highlight the importance of collaboration between different sectors to address the challenges related to rabies. Through this collaboration, comprehensive strategies can be developed to meet the needs of affected communities, overcoming limitations imposed by the lack of resources and the infrastructure of affected communities [26,27,28].

In conclusion, this research study sheds light on the importance of adopting a participatory and collaborative approach to tackle the health challenges faced by communities. By prioritizing proactive and personalized interventions, implementing awareness campaigns, and ensuring access to health and education, public health organizations can make substantial progress in the control and prevention of rabies in rural and vulnerable areas. The lessons learned from this research contribute to the growing body of knowledge in the field of health education and disease prevention, emphasizing the importance of continuous efforts and collaboration in combating infectious diseases like rabies.

## 5. Conclusions

In conclusion, this article sheds light on the urgent need for comprehensive measures to address the challenges posed by rabies in rural communities and regions with a low Human Development Index. The findings emphasize the importance of continuous education and awareness campaigns targeting both healthcare professionals and individuals at higher risk of rabies exposure. By enhancing the knowledge of the disease, its transmission, and preventive measures, we can empower communities to take proactive steps in protecting themselves and reducing the incidence of rabies.

Moreover, collaboration among different sectors, including healthcare, education, and public health sectors, is crucial for the successful implementation of preventive strategies. By working together, these stakeholders can develop integrated approaches that consider the unique circumstances and challenges faced by each community. This collaboration should extend beyond the healthcare sector to include community leaders, educators, and policymakers, as their involvement is essential in creating sustainable and effective interventions.

By prioritizing education, strengthening public health policies, and fostering multidisciplinary collaboration, we can strive towards a future where rabies is better controlled and the impact on affected communities is minimized. The lessons learned from experiences such as the one presented here serve as a reminder of the importance of continuous efforts in raising awareness, improving access to healthcare services, and empowering individuals to take an active role in their own health. 

Suggested strategies include establishing a multidisciplinary task force, allocating resources for vaccine procurement and healthcare professional training, empowering community leaders, and actively engaging communities. Regular monitoring and evaluation are necessary to adjust strategies as needed and share best practices among the affected communities. Through these collective actions, we can work towards the prevention and eventual elimination of rabies, ultimately contributing to the well-being and safety of communities worldwide.

## Data Availability

The data analyzed in this study are not publicly available due to privacy and ethical restrictions. The dataset used for this research contains sensitive patient information and is subject to confidentiality agreements. Therefore, the data cannot be shared openly. However, researchers interested in accessing the data can contact the corresponding author and request access, subject to the approval of the ethics committee and compliance with relevant data protection regulations.

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
