# Peer review of "Empowering Riverine Communities in the Amazon: Strategies for Preventing Rabies"

_ijerph, 2024, doi:10.3390/ijerph21010117_

Round 1

Reviewer 1 Report

Comments and Suggestions for Authors

This manuscript presented, it becomes evident that rabies represents a significant public health challenge, especially in rural areas Amazon and regions with lower Human Development Index.

When I read the title of this article, I got the impression that it could be a research article. However, this article was written as like a review article.

Although the article is very well written, important issues such as what its methodology is, what material is used, and the number of subjects are not clear.

The title and content do not match.

It is not stated how many people have been vaccinated.

It is not reported how many people were bitten after being vaccinated.

There is not enough evidence that vaccination and education training reduce the risk of Rabies infection.

Comments on the Quality of English Language

It should be corrected some words. Please use general terminology. Not only dog! "carnivore" or "domestic and wild carnivores".

Author Response

I appreciate the careful review of the manuscript. I acknowledge the concern raised regarding the apparent disconnect between the title and the content of the article. I have made a modification to the title to align it more closely with the study's proposition.

Regarding the methodology, I have added details to provide clarity, such as the number of people vaccinated, which was conducted only with the 30 individuals who experienced bat attacks and were subsequently interviewed. Additionally, I included information in the methodology section about the population size in the communities, the territorial dimensions, and the distance from the urban zone, illustrating the challenges in accessing information and assistance. Unfortunately, we do not have information on the incidence of bites after vaccination to provide more detailed data, as these communities are difficult to access due to dispersed residences over a vast territorial expanse, and the lack of river traffic throughout the year due to drought and river recession.

I appreciate your feedback on the insufficient evidence regarding the effectiveness of vaccination and education in reducing the risk of rabies infection. I have revisited this section, incorporating some relevant references to underscore the importance of these actions without emphasizing that they are sufficient for reducing cases. The word 'cachorro' was also corrected to 'domestic and wild carnivores.

Your insights are valuable and significantly contribute to improving the quality and clarity of the article. I am committed to making the necessary modifications to address these issues effectively

Reviewer 2 Report

Comments and Suggestions for Authors

The abstract of the manuscript presents valuable findings addressing the critical issue of rabies transmission from bats in the Amazon region and its impact on impoverished communities. However, several shortcomings have been identified. Firstly, the abstract lacks specific data on the scale of the study and the number of individuals involved making it challenging to assess the study's overall significance. Secondly, the abstract does not provide information on the duration of the study, which is essential for understanding the sustainability of the interventions. Additionally, it does not mention the methodology used, making it difficult to evaluate the rigor of the research.

The introduction of the manuscript provides a comprehensive overview of the critical issue of rabies, particularly in the context of the Amazon region, and highlights the evolving role of bats in transmitting the disease. It references relevant research and local experiences, demonstrating a strong foundation in the existing literature. However, it lacks a clear statement of the research objectives or a concise summary of the article's structure, which would help readers understand the manuscript's focus.

The methodology could benefit from a more explicit description of the research design, including how the selection of the three specific communities (Curipatá, Pedra Branca, and Santi) was made and whether there might be selection bias in these choices. Additionally, the manuscript lacks information about the size and demographics of the study population, which is important for assessing the generalizability of the results. Second, while the use of mixed methods for data collection and analysis is a strength, it would be helpful to see more details regarding the specific instruments used for data collection (questionnaires, interview guides) to ensure transparency and replicability.

The results section of the manuscript describes the outcomes of the interventions conducted in the riverine communities to address hematophagous bat attacks and improve public health. It is evident that the proactive approach of prioritizing vaccination and conducting educational actions had a positive impact on the communities. The high level of residents' adherence to vaccination and their improved knowledge of preventive measures are noteworthy. However, several shortcomings in the presentation of results should be addressed. The manuscript lacks specific quantitative data or statistical analyses to quantify the effectiveness of the interventions. It would have been beneficial to see concrete figures regarding vaccination coverage rates, changes in knowledge levels, and the reduction in new cases of hematophagous bat attacks. Additionally, while challenges and limitations are briefly mentioned, a more comprehensive discussion of these issues and their impact on the results is needed to provide a more in-depth analysis. Overall, while the results indicate promising outcomes, more quantitative data and a thorough discussion of limitations would strengthen the validity and comprehensiveness of the findings.

The conclusion of the article effectively underscores the urgent need for comprehensive measures to address rabies in rural communities with low Human Development Index. It emphasizes the importance of education and awareness campaigns, as well as multi-sectoral collaboration to combat this public health challenge. However, the conclusion could be strengthened by providing more specific and actionable recommendations for policymakers, healthcare professionals, and community leaders. It would be beneficial to outline a roadmap for the implementation of these comprehensive measures, including timelines and resources required. This would provide a more realistic and holistic perspective on the challenges and opportunities in this critical area of public health.

Comments on the Quality of English Language

Minor editing of the English language is required.

Author Response

Thank you for the detailed analysis of the manuscript and the constructive feedback. I will address each point raised:

Lack of Specific Data in the Abstract:

I understand the importance of providing more specific information about the study's scale, the number of individuals involved, and the duration of the research. I have made adjustments to the abstract to include these crucial details, offering a more comprehensive view of the study.

Absence of Clear Objectives in the Introduction:

I agree that a clear statement of the research objectives in the introduction is fundamental. I have revised the objective, providing clearer guidance to readers on the manuscript's focus.

Methodology Improvements:

I addressed information about the selection of communities, the demographics of the studied population, and the topics covered in health education. I highlighted that 100% of the 30 individuals interviewed reported being unaware of the disease and preventive measures, recognizing the importance of these actions in empowering the community to become information multipliers and proactive in preventing this health issue.

Deficiencies in Results Presentation:

I acknowledge the need for specific quantitative data and statistical analyses to substantiate the effectiveness of interventions. However, due to the small sample size as a limitation, statistical data was not included, emphasizing the significance of educational efforts that can empower individuals in remote areas without access to health services for rabies prevention. It is emphasized that these communities are located in a reserve in the heart of the Amazon, facing various limiting factors, not only health-related, and currently experiencing a historic drought, leaving them isolated. Educational activities are essential in this context of vulnerability.

Conclusion:

I have added more specific and practical recommendations for policymakers, healthcare professionals, and community leaders for the implementation of comprehensive measures, providing a more realistic and action-oriented perspective.

I appreciate the valuable guidance and am committed to refining the manuscript based on these considerations.

Reviewer 3 Report

Comments and Suggestions for Authors

Develop coherence and flow in describing the materials and methods. At first, it should be mentioned in the methodology how many interviewers & interviewee were involved, and also elaborate the approach of interview, how it was conducted, what particular questions were they asked to get the understanding of their knowledge about the prevention of bat bite and its significance. Moreover, need correction in line 285.

Comments on the Quality of English Language

Over usage of comma to break the long sentences, instead of it use the complete short sentences

Author Response

Thank you for reviewing the manuscript. To enhance coherence and flow in the description of materials and methods, I made adjustments to clarify the approach. I acknowledge the need for specific quantitative data and statistical analyses to substantiate the effectiveness of interventions. However, the total population of the communities is 345 people, and we only interviewed those who had experienced bat attacks due to the small sample size, which was 30 individuals. Due to the sample limitation, statistical data was not included, emphasizing the importance of educational efforts that can empower individuals in remote areas without access to health services for rabies prevention. It is noteworthy that these communities are located in a reserve in the heart of the Amazon, facing various limiting factors, not only health-related, and currently experiencing a historic drought, leaving them isolated. Educational activities are essential in this context of vulnerability. Additionally, I corrected line 285 to ensure accuracy and clarity in that section of the text. I hope these modifications contribute to a more consistent and understandable presentation of the methodological process. If there are more specific points that deserve attention, I am open to additional suggestions to further improve the manuscript.

Round 2

Reviewer 1 Report

Comments and Suggestions for Authors

The researchers generally answered my questions. I believe that answering the questions I have stated below will make the subject more understandable.

The mere use of the word bat in the title creates the impression that it has nothing to do with Canine Rabies infection. Is the infection in the area only from contact with bats?

Were these 30 people vaccinated voluntarily?

We know that curative vaccination is usually performed in humans against rabies. Human vaccination for prophylaxis against rabies is not routine.

Author Response

Thank you for your considerations on the manuscript.

1)Regarding the simple use of the word 'bat' in the title, does it suggest that Canine Rabies infection is related only to contact with bats?

Response: The revised title, 'Empowering Riverine Communities in the Amazon: Strategies for Preventing Rabies,' has been adjusted to emphasize a more comprehensive approach to rabies prevention. However, it is worth noting that infection in the area primarily occurs through contact with bats, and the title aims to address general prevention strategies.

2)Were the mentioned 30 individuals vaccinated voluntarily?

Response: Yes, it is important to note that vaccination was carried out voluntarily by individuals who chose to participate in the vaccination program. This was highlighted in lines 156 and 157 of the text.

3)We know that curative vaccination is usually performed in humans against rabies. Human vaccination for prophylaxis against rabies is not routine.

Response: Indeed, in the reserve area, located in the heart of the Amazon Rainforest without regular health services, human vaccination for prophylaxis against rabies is not carried out as a routine practice. The absence of infrastructure and the remote location impact the implementation of preventive vaccination practices.

Reviewer 2 Report

Comments and Suggestions for Authors

The authors have addressed the suggestions and shortcomings. The manuscript has been improved substantially. It is my pleasure to recommend the manuscript for publication in its present form.  

Author Response

Thank you for your considerations on the manuscript.